# The Use of Subsidence to Estimate Carbon Loss from Deforested and Drained Tropical Peatlands in Indonesia

**Gusti Z. Anshari** [1,2] 🔟**, Evi Gusmayanti** [1,3] **and Nisa Novita** [4,*]

1   Magister of Environmental Science, Universitas Tanjungpura, Pontianak 78124, Indonesia; gzanshari@live.untan.ac.id (G.Z.A.); evi.gusmayanti@faperta.untan.ac.id (E.G.)
2   Soil Science Department, Universitas Tanjungpura, Pontianak 78124, Indonesia
3   Agrotechnology Department, Universitas Tanjungpura, Pontianak 78124, Indonesia
4   Yayasan Konservasi Alam Nusantara, DKI Jakarta 12160, Indonesia
*   Correspondence: nisa.novita@tnc.org

**Abstract:** Drainage is a major means of the conversion of tropical peat forests into agriculture. Accordingly, drained peat becomes a large source of carbon. However, the amount of carbon (C) loss from drained peats is not simply measured. The current C loss estimate is usually based on a single proxy of the groundwater table, spatially and temporally dynamic. The relation between groundwater table and C emission is commonly not linear because of the complex natures of heterotrophic carbon emission. Peatland drainage or lowering groundwater table provides plenty of oxygen into the upper layer of peat above the water table, where microbial activity becomes active. Consequently, lowering the water table escalates subsidence that causes physical changes of organic matter (OM) and carbon emission due to microbial oxidation. This paper reviews peat bulk density (BD), total organic carbon (TOC) content, and subsidence rate of tropical peat forest and drained peat. Data of BD, TOC, and subsidence were derived from published and unpublished sources. We found that BD is generally higher in the top surface layer in drained peat than in the undrained peat. TOC values in both drained and undrained are lower in the top and higher in the bottom layer. To estimate carbon emission from the top layer (0–50 cm) in drained peats, we use BD value 0.12 to 0.15 g cm$^{-3}$, TOC value of 50%, and a 60% conservatively oxidative correction factor. The average peat subsidence is 3.9 cm yr$^{-1}$. The range of subsidence rate per year is between 2 and 6 cm, which results in estimated emission between 30 and 90 t CO$_2$e ha$^{-1}$ yr$^{-1}$. This estimate is comparable to those of other studies and Tier 1 emission factor of the 2013 IPCC GHG Inventory on Wetlands. We argue that subsidence is a practical approach to estimate carbon emission from drained tropical peat is more applicable than the use of groundwater table.

**Keywords:** bulk density; carbon loss; drained peat; organic carbon; peat forest; subsidence

## 1. Introduction

Carbon loss in drained peat occurs globally, particularly in tropical regions. Drainage typically occurs during peat swamp forest conversion to agricultural lands, primarily for oil palm and acacia plantations. Drainage involves lowering the groundwater table, thus providing an aerobic environment for the cultivation of non-native plant species of the tropical peat. Consequently, the peat decomposition rate accelerates due to microbial oxidation of organic matters (OMs) in drained peats [1–5]. Currently, the measurement of the groundwater table is the most acceptable model for estimating carbon loss in drained peat [4,6–10]. However, research has shown that the groundwater table does not always positively correlate with carbon emission [11,12]. Other factors, such as soil moisture, soil pH, and fertilizer application [6,13–15], are significant predictors in estimating carbon loss from drained peat soil. Unfortunately, continuous monitoring of groundwater table and carbon emission is not practical in the tropical peatlands in Southeast Asia due to expensive automatic chambers, lack of electrical power in remote peat area, and lack of skilled human

resources on peatlands carbon accounting. Therefore, there is a need to measure carbon loss that can practically and accurately be implemented in tropical peatlands in Southeast Asia.

We propose a robust and straightforward approach to estimate carbon loss in drained peat, using a yearly subsidence rate. This approach is more practical than using a ground-water table as a predictor, which is naturally dynamic and needs to be continuously monitored. The installation of a peat subsidence monitoring tool is relatively easy, using a sturdy subsidence pole implanted to mineral substratum underneath peat. The measurement of subsidence is monitored every three or six months. The material for constructing a subsidence pole is cheap and widely available, i.e., using strong metal or high-quality PVC poles.

## 2. Methods

We collected both published and unpublished data of BD and TOC from drained and undrained peats. We used published subsidence rate data. Locations of these studies cover both coastal and inland peats in Kalimantan, including Sarawak, northern Borneo, and the eastern coast of Sumatra. Bulk density and TOC data representing undrained peats are derived from Lake Siawan, West Kalimantan; Maludam National Park, northern Borneo; Sebangau National Park, Central Kalimantan; and Batang Hari River, Jambi Province, Sumatra. Bulk density and TOC data representing drained peats are derived from Rasau Jaya peat complex, West Kalimantan; Jabiren, Central Kalimantan; and Tanjung Jabung Barat, Jambi. We compiled a big data set of BD and TOC in Table S1. In addition, data subsidence is derived from published studies in Riau, Jambi, West Kalimantan, Central Kalimantan, Aceh, and Western Johor. Figure 1 shows research sites where bulk density (BD), total organic carbon (TOC), and subsidence rates were compiled and analyzed.

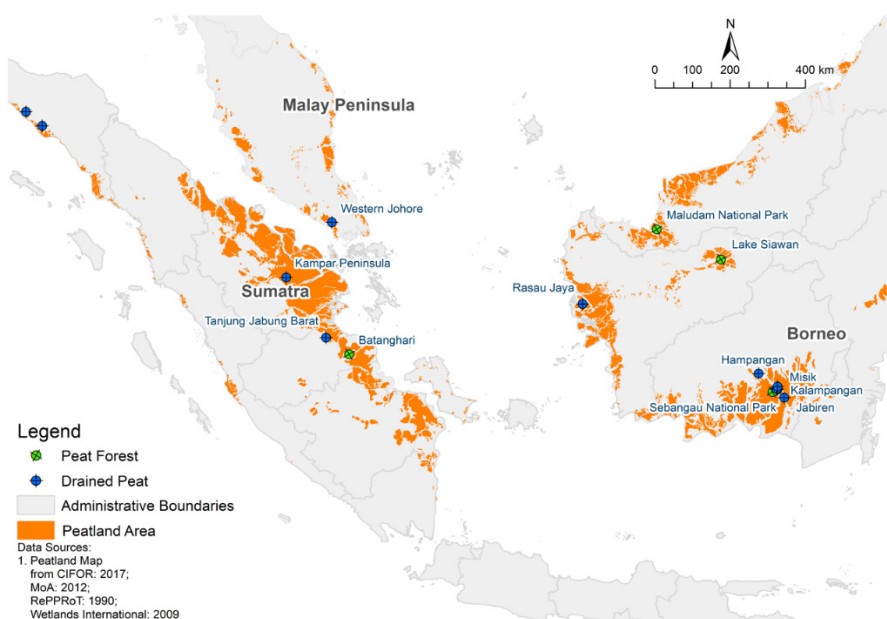

**Figure 1.** Locations of drained peats for agriculture and undrained peat forests where data on bulk density (BD), total organic carbon (TOC), and subsidence rates were compiled and analyzed in this review. Please note that status of peat forests is presently subject to change to drained peats, i.e., Batang Hari, Jambi and Sebangau National Park, Central Kalimantan.

Sample depth intervals used to measure BD and TOC varied from one study to another. We regroup the sample depth at 50 cm intervals to compare these data from one study to another. We analyzed 308 samples of drained peat, and these data cover the top surface 50 cm until 200 cm depth. We analyzed 259 samples in the undrained peat forest, covering up to 300 cm depth. Profiles of BD and TOC are pr Pearson correlation and scatter plot are

utilized to explain the correlation between BD and TOC in the drained and deforested site and the undrained peat forest.

Despite the variation of subsidence rate data, we compiled subsidence data post ten years of canal construction. We aim to analyze only the subsidence rate in drained peats, which have been converted into agricultures since more than ten years ago. Land uses covering subsidence data are Acacia plantation, oil palm, rubber plantation, and other cultivated crops. Based on BD, TOC, and subsidence rate, we calculate carbon (C) loss due to drainage and land-use change (See Equation (1) in Section 8).

## 3. Distribution, Vegetation Formation and Uniqueness of Tropical Peat

Peat occurs in all climatic zones, from tropic to arctic [16], and the area of total global peat is approximately 423.2 million ha [17]. Food and Agriculture Organization [18] estimated the range of global peat is between 325 to 375 million ha. However, these peats were primarily found in the boreal, sub-artic, and arctic regions of the Northern Hemisphere, temperate region of Western Europe, northern Scandinavia, and West Siberia. Some peats may also be found in the cool mountain [18–20]. Only about 10% of peats are located in tropical regions of South America, Africa, and Southeast Asia [18,21–24]. The tropical peats in Southeast Asia were initially determined to be the largest, about 25 million ha (Mha), and were mostly located in Sumatra, Kalimantan, and Papua [22,23,25]. Gumbricht et al. [26] challenged the extent of tropical peat. They suggested that the most extensive tropical peat area is located in America, up to 63 Mha, and the total tropical peat was estimated to be 150 Mha instead. Conservative estimates of tropical peats range from 36 to 47 Mha [21,22].

The tropical peat forest ecosystem is special and unique. Tree species in tropical peat forests have some similarities to tree species in dipterocarp forest on drylands. Most tree species have a particular organ to cope with an inundated environment. Many trees in this tropical swamp have pneumatophores [27]. In contrast, other trees uniquely adapt to high groundwater tables by growing on the raised peat surface (hummocks or ridge), creating aerobic and fertile rooting zone [28]. The lower peat surface is called slough or hollows, which are commonly water-logged. Lampela et al. [27] conclude that the formation of microtopography of tropical peat is through random processes. It was hypothesized that the varying accretion rates of OMs in ridge and slough play a significant role in forming the undulating surface microtopography. Large trees are predominantly located on the ridge (hummocks), which provide the trees with the added benefit of more nutrients and oxygen than in the slough [28,29]. The availability of oxygen in hummocks supports the growth of plants and speeds the decomposition of OMs [29]. Large trees tend to occupy the ridges, while other adapting water-logged vegetation tends to occupy the slough [27].

Organic matters that form tropical peats are derived from mangrove trees, freshwater swamp trees, heath forests, grass, and sedge ferns. A variety of vegetation formations occurs in a tropical peat swamp in Southeast Asia. For example, Anderson [30] and Anderson and Muller [31] described the six vegetation formations in the coastal peat dome in Northern Sarawak. These vegetation formations in tropical peat domes show unique adaptation capabilities of peat swamp vegetation to changes of peat thickness. Tall and small-diameter trees in the central peat dome area better adapt to the less fertile peat than the mixed and large trees in the shallow and regularly inundated peat area. However, episodic dry spells associated with El Niño may cause a substantial drop of water storage in the peat dome, leading to a drop of water storage in the central peat dome. This explains why the stunted and low pole forest is predominantly in the center of the peat dome. The colonization of Nephentes is common in the peat region, which is deficient in nitrogen and frequently suffers from water stress.

In reality, the vegetation zonation in every peat dome seems to adapt to local conditions. Page et al. [32] report the diversity of vegetation formations in Sebangau peat dome, Central Kalimantan. Low pole and tall interior forests are typically found in the deep peat (8–10 m). Mixed dipterocarp forests occur in shallow peat (2–3 m). The riverine forests

occur in the very shallow peat area (about 1–1.5 m). In Sumatra, Kuniyasu and Tetsuya [33] described some similarities between the vegetation associations in tropical peat forests in the riparian area and the tidal and deltaic peats in the Kerumutan River, eastern Sumatra.

Tropical peat swamp contains rich and diverse flora, yet they are not well valued. Lucrative timber species such as Shorea blangeran, Dyera polyphilla, and Gonystylus bancanus are intensively logged, causing deforestation and habitat destruction. The removal of vegetation in the tropical peat swamp forest causes the depletion of OMs, which are necessary for maintaining peat accumulation. In addition to a continuous supply of OMs, the waterlogged environment that controls the rate of OM decays is a prerequisite for peat formation in the tropic. High percentages of aromatic chemicals in tropical peats may inhibit the rapid decomposition rate in a warm climate [21,34]. Peat accumulation does not occur when the supply of OM input is either intermittent or limited, which leads to a higher decomposition rate of dead OMs compared to the delivery rate of freshly dead OM to the peat-forming site. Therefore, under anthropogenic disturbances, such as deforestation, drainage, and fires, peat accumulation stops [35] and, consequently, peats act as a source of carbon to the environment [36–38].

## 4. Selected Peat Properties Affecting Carbon Stock

Based on soil taxonomy, peat is commonly known as either Histosols [18,39], or Organosols [40]. The content of elemental, organic carbon in OMs determines physical, chemical, and biological properties of Histosols. According to Soil Survey Staff [39], the elemental organic carbon in Histosols is greater than 12%. The thickness of OMs is minimum in the range of 40 cm to 60 cm, depending on the rate of peat decomposition. Fibric peat must have at least 60 cm thick of continuous OMs, and both hemic and sapric peats must have 40 cm thick OMs. In Indonesia, the minimum thickness of OMs is 50 cm [40]. Peaty soil has less than 50 cm of OMs. The amount of OMs in tropical peat soil range between 65% and 99% [23,41–43]. This OM content plays a vital role in water storage. Under extreme dryness conditions, OM is not capable of absorbing water (hydrophobic). Healthy peat is hydrophilic, permanently bonding with water through hydrogen bonds (carboxyl and hydroxyl groups). Hydrophobic peat has more non-polar aromatic compounds than polar aliphatic compounds [12]. The loss of water retention capability of peat soil results in hydrophobic peat, particularly the acrotelm layer, which makes the peat soil susceptible to fire [12,44]. Furthermore, fibric and hemic peats are less likely to be stable than sapric peat due to the decomposition of aliphatic organic compounds in newly deposited OMs [45].

Bulk density of tropical peat is very low, ranging from $0.04 \text{ g cm}^{-3}$ to $0.30 \text{ g cm}^{-3}$ [21,43,46]. The average bulk density values of fibric, hemic, and sapric peats are 0.09 g cm-3, $0.12 \text{ g cm}^{-3}$, and $0.20 \text{ g cm}^{-3}$. An increase in bulk density of peat soil is irreversible due to physical compaction and consolidation of OMs. Drainage significantly causes an increase in bulk density and is associated with the reduction of elemental organic carbon content [47]. An increase of $0.01 \text{ g cm}^{-3}$ in bulk density causes a decrease of $0.6 \text{ mg C g}^{-1}$ [48]. Therefore, an increase in bulk density indicates peat subsidence, which is directly associated with carbon emission to the atmosphere [1,4,8,10], and water discharge [49–52].

## 5. Peat Degradation

Deforestation reduces the supply of OM, a parent material for peat formation. The removal of large commercial trees due to logging causes the collapse of ridges, where large trees commonly live. After logging, the logged-over peat swamp forest may suffer from vegetation degradation and the change in surface microtopography that regulate the spatial distribution of vegetation assemblages in tropical peat swamp forests. Moreover, timber harvesting has significant impacts on the continuity of OM delivery, consequently inhibiting the development and growth of native vegetation in tropical peat swamp forests. The initial process that leads to the development of native vegetation into several vegetation assemblages under deficient nutrients and an unfavorable environment in the peat swamp

forest is currently unknown. In general, vegetation formations in the tropical peat habitat are circular and singular, showing adaptation to one peat habitat relative to another peat habitat [53]. Figure 2 illustrates anthropogenic disturbances that cause peat subsidence in tropical peat swamp forests. Subsidence is globally considered as a major criterium of peat degradation in drained peatland.

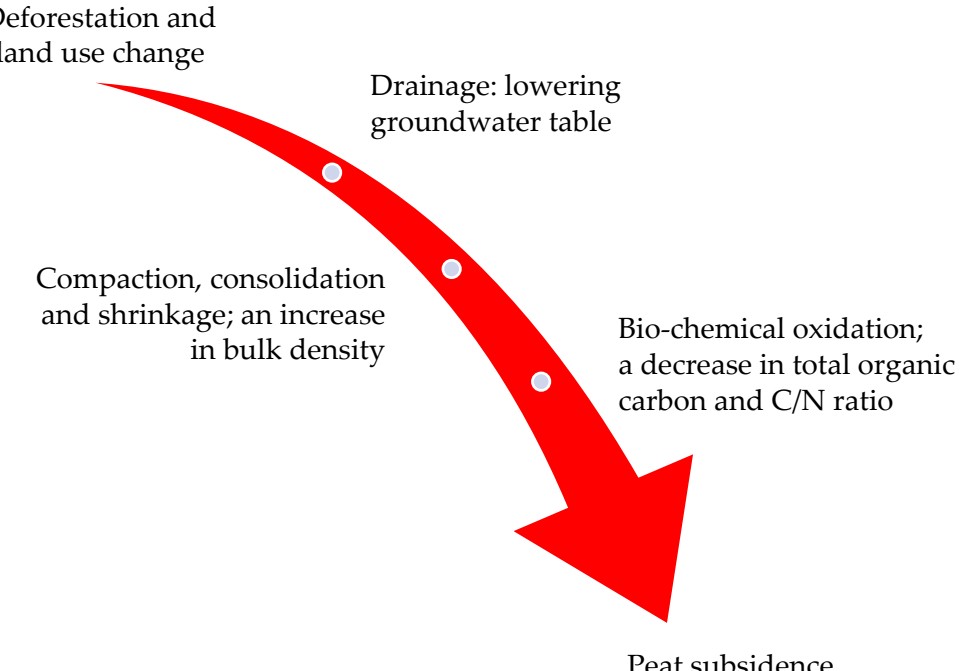

Deforestation and land use change

Drainage: lowering groundwater table

Compaction, consolidation and shrinkage; an increase in bulk density

Bio-chemical oxidation; a decrease in total organic carbon and C/N ratio

Peat subsidence

**Figure 2.** The relationship between anthropogenic disturbances on tropical peat forest ecosystem and peat subsidence.

Drainage causes severe degradation because of the change of the hydrological cycle and hence water balance. In a natural state, peatland acts as an important wetland that stores water. Drainage leads to groundwater table depletion, withdrawn by an increase in water discharge and evaporation from the open canopy or deforested peat. Anthropo-genically drained peat leads to rapid subsidence and the change of peat properties. Drainage triggers rapid decomposition, which is caused by the breakdown of phenolic compounds under aerobic conditions [54,55]. The significant impact of drainage on the loss of peat surface or subsidence in drained peat is widely recorded [1,2,8,56–62].

## 6. Subsidence

In a natural state, the groundwater table in peat is always high. The primary factors that determine water storage in peats are rainfall, evapotranspiration, and water discharge. When considering the hydrological property of peatlands, their thickness, decomposition, type of peat landscape, land cover, hydraulic conductivity, and drainage canal are some of the most relevant factors that control the property. Moreover, depending on the location, river tides control the groundwater table in coastal peat. Meanwhile, forested land cover on tropical peat forests protects direct exposure of peat surfaces from sunlight and controls the rate of evaporation. During a period of seasonal drought, the groundwater table significantly declines, but the peat surface is still able to maintain its moisture and stays wet. Drained and deforested peat suffers from water deficit and loss of particle of OM, which makes peatlands prone to irreversible dryness. Furthermore, lowering the groundwater table increases oxygen availability for peat decomposition, hence increasing carbon emission. Consequently, drained peats suffer from water deficit during droughts and are, therefore, easily ignited, which leads to more carbon emissions.

Peat under anthropogenically drainage disturbance always subsides due to both physical and chemical changes of OMs. Therefore, compaction and consolidation of

particles in OMs are direct consequences of anthropogenic disturbances on peat soil. This change leads to pore space reduction and bulk density increase. Moreover, oxidation causes a decline of carbon concentration in the peat matrix, and this chemical removal process in the peat surface contributes to carbon release. The rate of peat subsidence is initially high in the first year of drainage, and then the rate stabilized in the following years. The average rate of peat subsidence in Southeast Asia is reported to be around 2.2 cm per year [63]. However, Evans et al. [64] reported that the average rate of peat subsidence in Acacia plantation on tropical peat in Sumatra might reach as high as 4.3 cm per year.

To combat the loss of carbon, managing a high water table and keeping peatland wet is the only practice that would decline the present peat subsidence by 20 to 30% [4,64,65]. Thermal subsidence due to fire is varied mainly due to differences in fire regimes and the capacity of water retentions in peat. Based on a remote sensing analysis, Khakim et al. [66] reported that the 2015 fire caused drained peat in South Sumatra to subside in the range of 12 cm to 250 cm, but this result still needs to be validated. Wösten et al. [8] propose that every 10 cm water drainage results in a 0.4 cm subsidence rate per year in tropical peat in Sarawak, Malaysia. However, drainage does not only cause groundwater table withdrawal, but also it causes changes in the physical, chemical, and biological properties of peat. An increase in bulk density is commonly observed post drainage as OMs shrunk and became more compact and consolidated. Therefore, Darmawan et al. [67] suggested that an increase in bulk density is not a good indicator to determine compaction because of the fluctuation of moisture contents. Seasonally, OMs swell during the wet season, and, in contrast, OMs shrink during the dry season. To take these characteristics into account, an average of bulk density in both seasons is considered appropriate to assess peat compaction based on an increase in bulk density value caused by drainage [68].

Researchers have also considered plant species as a predictor for carbon emission. However, this factor is not preferred because, in a natural state, tropical peat forests consist of a variety of vegetation formations. Microbial activities control rates of peat decomposition, while the vegetation functions as a supplier of OMs. Carlson et al. [69] reviewed a positive relation between long-term groundwater table (~20–110 cm) and carbon loss from plantations on drained tropical peat. The model suggests that an average groundwater table of 70 cm would result in total emission in the range 18 to 22 t C ha$^{-1}$ yr$^{-1}$, with an average of 20 t C ha$^{-1}$ yr$^{-1}$. This model uses a subsidence approach to estimate the amount of carbon emission. For comparison, the IPCC default values of total emission from oil palm and acacia plantations on peat are 11 t C ha$^{-1}$ yr$^{-1}$ and 20 t C ha$^{-1}$ yr$^{-1}$ [70], respectively. Carlson et al. [69] concluded that vegetation species do not significantly affect the total emission and suggest that total carbon emissions from oil palm (Elaeis guineensis) and acacia timber (Acacia crassicarpa) plantations are indifferent. However, this conclusion does not yet cover vegetable and horticultural agricultures on tropical peatland. Jamaludin et al. [71] reported a large carbon emission from small-scale agriculture from drained peats in West Kalimantan. Khasanah and Noordwijk [72] and Wakhid et al., [56] also reported a large carbon emission from small-scale agricultures of rubber (Hevea brasiliensis), coffee (Coffea liberica), betel nut (Areca catechu), and mixed coconut (Cocos nucifera) garden. Nurzakiah et al. [12] reported relatively low carbon emission from rubber plantation intercropping with pineapple, which was 6 t C ha$^{-1}$ yr$^{-1}$, and from traditional rubber plantations, which was 5 t C ha$^{-1}$ yr$^{-1}$. Low emission values in the agricultural land can be attributed to the natural hydrophobicity of selected aromatic organic compounds such as lignin [21], in that when the tropical peat soil particles are not dissolved in water and the particles become recalcitrant for biological decomposition [15,73]. This finding seems to confirm the conclusion that plant species are not a good predictor for estimating carbon emission from drained peats in the tropic.

Other factors that influence carbon emission from drained tropical peats are rarely investigated. These include OM decomposability, surface microtopography, fertilizer application, soil moisture, and soil and air temperatures. Knowledge of chemical variations that affect carbon emission in tropical peat is very important for understanding how carbon

emits from peat soil [74]. However, the chemical composition of OMs in tropical peat is very diverse as the origin of tropical peat is derived from various swamp tree species, which makes chemical analysis challenging.

The micro-topography of the peat surface is primarily composed of hummocks and hollows [28,75]. Hummocks play a significant role in providing a rooting zone, a source of total $CO_2$ emission. Either regular or permanent inundation characterizes hollows or depression lenses. Irregular patterns of hummocks and hollows in tropical peat surfaces influence the unique distribution of tree diversity [28]. Anaerobic condition supports methanogen activity to produce $CH_4$ under water-saturated peat surface [76]. In general, depending on groundwater table depth, the main source of carbon emission is from 0–30 cm peat surface [74].

Degradation of tropical peat forests also occurs in Asia, Africa, and America. The degree of tropical peat degradation is larger in Southeast Asia than those in Africa and America. Factors that control this degradation are remoteness of the location, human population density, law enforcement, and government policies on the protection of tropical peats [77]. Overall, anthropogenic drainage leads to the degradation of tropical peat forests, which change the ecological function from a carbon sink into a source [37,62,78–80]. Peats degradation around the world is predicted to act as a carbon source under global warming [37,81–83]. Figure 3 shows a PVC subsidence pole and anthropogenic disturbances on tropical peats.

(a) (b) (c)

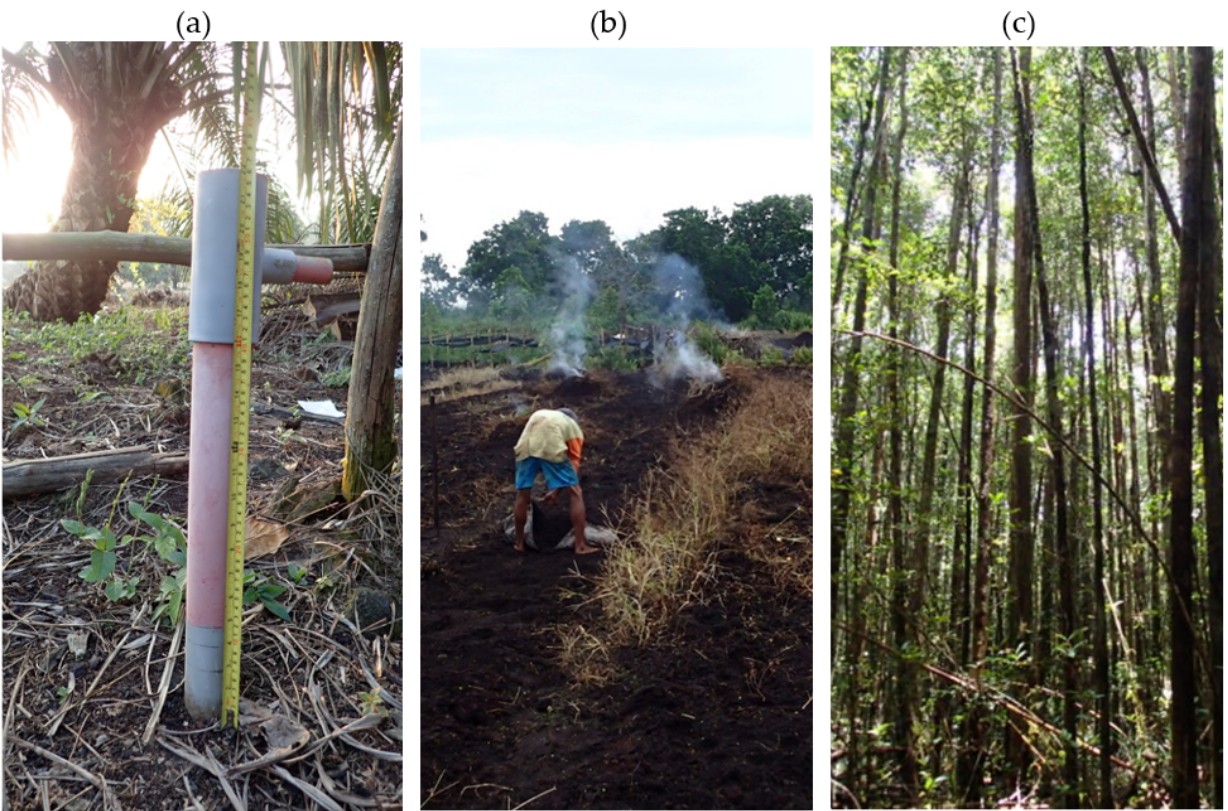

**Figure 3.** A PVC (Polyvinyl chloride) subsidence pole at an oil palm plantation (**a**); Slash and burnt farming on peat (**b**); and a logged-over- peat forest in the upper Kapuas River, West Kalimantan Province, Indonesia (**c**). Photos: Gusti Anshari.

Drainage provides ample oxygen into the acrotelm, which is an upper layer above the groundwater table. The removal of water in this layer causes OMs to shrink, compact, and consolidate. As a result, most pore spaces are filled with oxygen, which provides a favorable environment for biological decomposition. Drainage directly erodes the height of the peat surface, which is commonly used as a good predictor of carbon loss. Peat surface subsidence is divided into primary and secondary subsidence. Compaction and consolidation are

initially led to physical changes, but both processes may reduce peat volume or shrinkage. However, this primary subsidence does not lead to carbon loss. Subsidence processes continue into secondary, which leads to oxidation of OMs or persistent carbon emission post drainage. The estimated oxidation caused by peat subsidence range from 60 to 75% [4,8,61,84]. The rate of peat subsidence is initially between 20 to 50 cm per year in the first or second year of drainage [8]. Then, the rate of peat subsidence reduces into 4 to 6 cm per year [4,8,64]. Finally, the rate of peat subsidence becomes stable at 1.5 to 2 cm per year [8,63]. It is estimated that post ten years of deforestation and drainage, the Sebangau peatland in Central Kalimantan suffered from total compaction in the range 2 to 4 m [68], causing a substantial reduction of peat depth.

Peat subsidence caused by fire is not easily studied as fire consumes both vegetation biomass and the dry fraction of OMs in peat. Peat fire is known as a smoldering fire that does not cause complete oxidation. Peat fires are commonly recurrent, emitting different amounts of carbon each fire event [85], and oxidation of OMs might continue post-fire occurrences [86]. Methane ($CH_4$) emission might increase if the post-fire groundwater table becomes sufficiently high and the peatland area is inundated [87].

The current model to estimate carbon loss from drained tropical peat uses a subsidence approach based on the groundwater table as a proxy of peat oxidation. Subsidence is divided into physical, biological, and thermal subsidence (See Table 1). Wösten et al. [8,59] firstly introduced this model, and the model becomes widely used and popular after the publication of Hooijer et al., [4,88]. Further, Wösten et al. [59] explained that the groundwater level at 40 cm from peat surface is critical for preventing drained peatland from fires. In sum, peat subsidence is unavoidable if a drainage canal is constructed.

**Table 1.** Subsidence and impact.

| Subsidence | Category | Impact |
| --- | --- | --- |
| Physical | Compaction, consolidation and shrinkage | Bulk density increase, and decrease in volumetric water content |
| Biological | Microbial oxidation/decomposition | Carbon emission/loss to atmosphere and water |
| Thermal | Smoldering fire | Haze pollution and carbon loss to atmosphere |

## 7. Profiles of Bulk Density and Total Organic Carbon

Peat thickness, BD, and TOC are important parameters to estimate carbon stock in peat soil. Carbon density (CD) is a product of BD times TOC. This sub-section presents distribution BD and TOC according to peat thickness and Pearson correlation between BD and TOC in the undrained peat forest and drained peatland.

Figure 4a shows that BD in the undrained peat forest is not statistically different at different peat thicknesses. The pattern shows a slight increase following an increase in peat thickness. The range of BD in the top layer (0–50 cm) is 0.09 to 0.13 g cm$^{-3}$. The overall mean BD in this site is 0.12 ± 0.06 g cm$^{-3}$. On the other hand, Figure 3b shows a declining pattern of BD following peat thickness in the drained peat. High BD in the top layer (0–50 cm) indicates peat compaction. The range of BD in this top layer is 0.12 to 0.15 g cm$^{-3}$, and the range of BD in the bottom layer (150–200 cm) is 0.08 to 0.09 g cm$^{-3}$. The overall mean BD in the drained peat is 0.11 ± 0.07 g cm$^{-3}$.

The pattern of TOC in Figure 5 shows a similar story to that of BD. Low TOC concentrations are observed in the top peat layer in both undrained and drained peat sites. The TOC values in drained and undrained peat are 49% to 52% and 50% to 52%, respectively. The range of TOC in the bottom layer in drained and undrained peat is 54% to 56% and 52% to 53%, respectively. These observations suggest that peat decomposition occurs in both drained and undrained peat, particularly in the top layer subject to the natural fluctuation of the groundwater table.

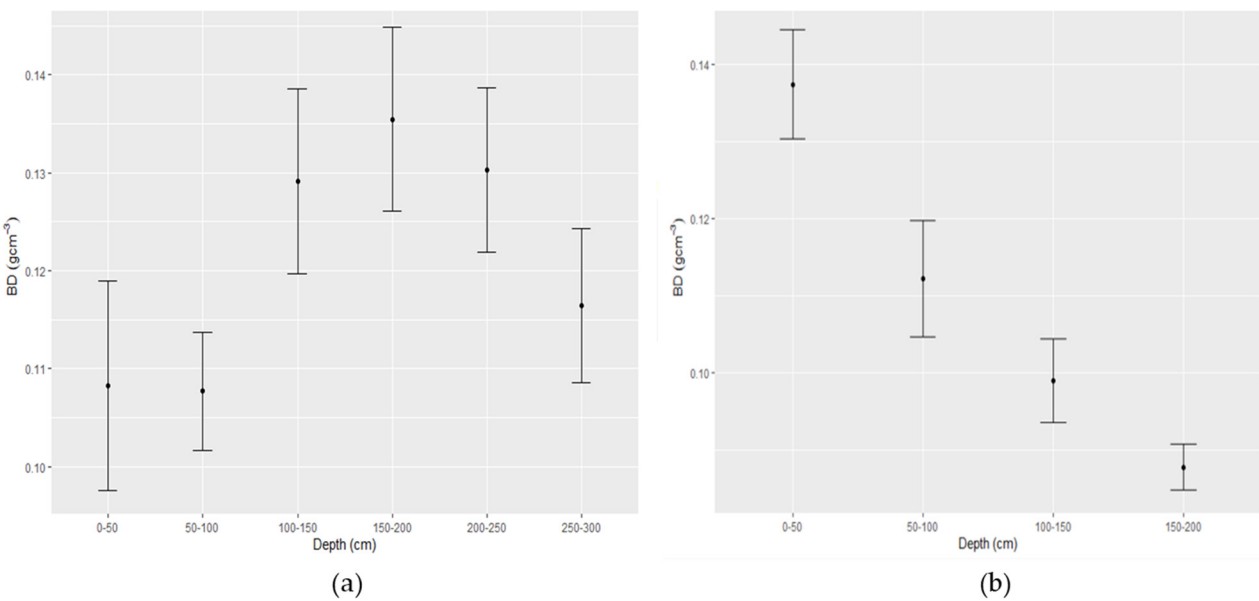

**Figure 4.** Mean plot ± SE of bulk density (BD) in the undrained peat forest (**a**) and in the drained peatland (**b**).

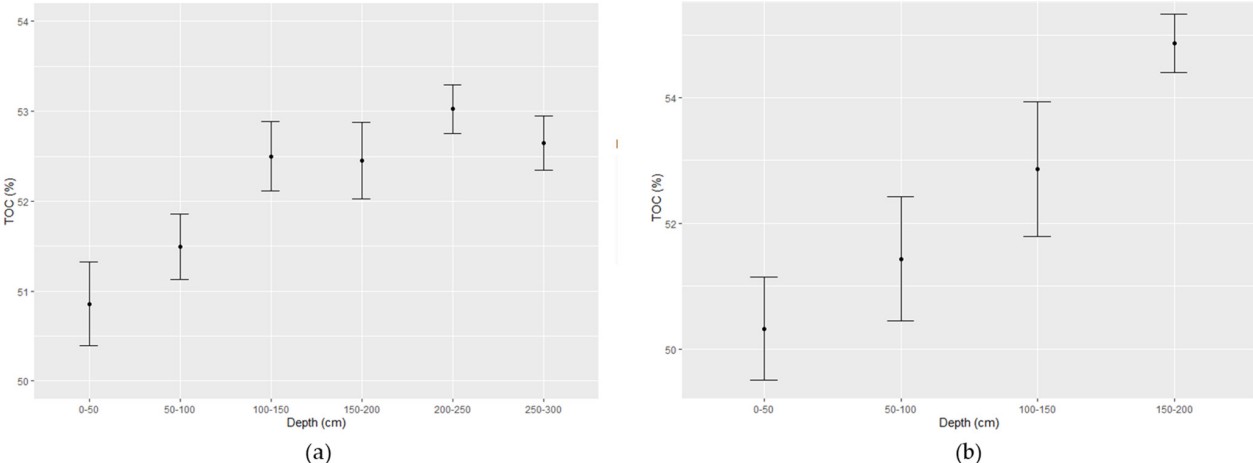

**Figure 5.** (**a**). Mean ± SE of total organic carbon (TOC) in the undrained peat forest, and (**b**). in the drained peatland.

Figure 6a,b show scatter plots between BD and TOC in the undrained and drained peats. Pearson correlations between BD and TOC in the undrained and drained peats are significant and become very strong in the drained peat. An increase in BD in the drained peat cause a significant decline in TOC, suggesting peat mineralization and carbon emission from deforested and drained peatland (See Figure 6b). Tonks et al. [48] reported a decline in TOC in relation to an increase in BD, indicating that TOC in drained and disturbed peat is not stable [89,90]. The stability of carbon in tropical peat is secured if only the peatland is kept saturated at all times [91–93].

It is worth noting the importance of peat compaction for improving the bearing capacity of peat soil and enhancement of capillary water for maintaining soil moisture in dry spells. Adhi et al. [94] reported that the average increase in BD in the top 10–20 cm peat used for oil palm plantation range from 0.12 to 0.15 g cm$^{-3}$, leading to the availability of capillary water throughout 30 cm peat surface. The average value of BD (0–50 cm) of peat planted with oil palm in Tanjung Puting, Central Kalimantan is 0.37 g cm$^{-3}$ [95]. Wakhid et al. [56] noted the average BD of 0.23 g cm$^{-3}$ in the top peat surface (0–75 cm) in the rubber plantation on peat. As shown in Figure 4b, high bulk density in the drained peat correlates with low carbon content or carbon loss. Tables S1 and S2 present BD and TOC in the undrained and drained peats.

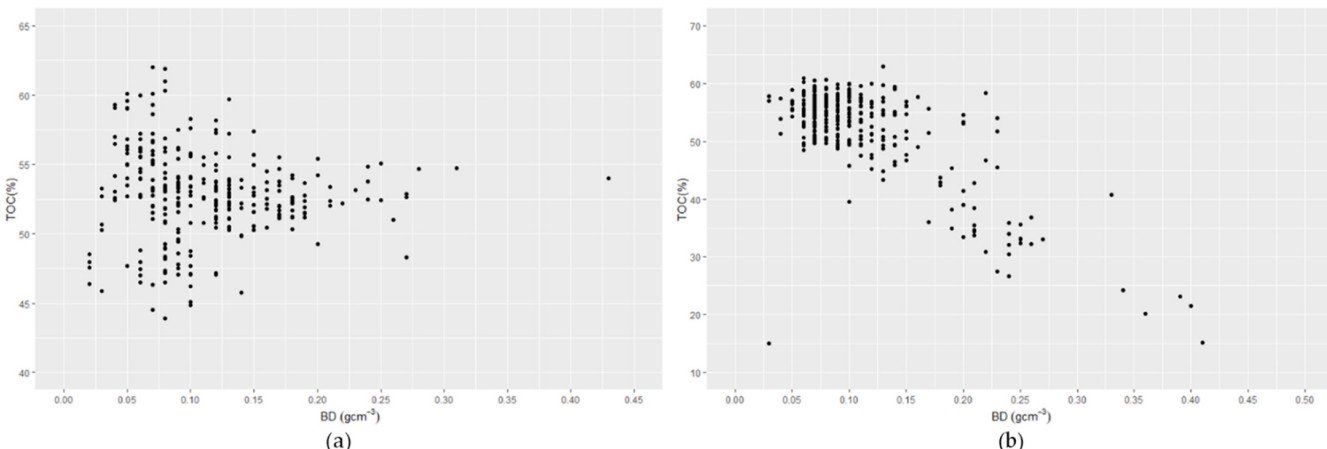

**Figure 6.** Scatter plots Bulk Density (g cm$^{-3}$) and Total Organic Carbon (%) showing (**a**) a low Pearson correlation (Pearson R = 0.15, $n$ = 259), in the undrained peat forest, and (**b**) a high negative Pearson correlation (Pearson R = −0.81, $n$ = 308) in the drained and deforested peat.

## 8. Carbon Loss Estimate

Bulk density (BD), TOC and peat subsidence (PS) can be used to estimate carbon loss in drained peat. Based on data shown in Figures 3 and 4, the mean BD in the top surface (0–50 cm) in the drained peat is 0.14 g cm$^{-3}$, and the mean BD value in the top surface (0–50 cm) in the peat forest is 0.11 g cm$^{-3}$. Mean TOC values in the top surface (0–50 cm) in both the drained and undrained peat forest is about 50%. Based on these values, the formula to estimate carbon loss due to an increase in bulk density caused by drainage and land use disturbance can be calculated using the following equation:

$$\mathrm{CL_{50}} = \mathrm{CD} \times \mathrm{PVL} \times 0.6 \times 3.66 \tag{1}$$

where:

$\mathrm{CL_{50}}$ = carbon loss (t $\mathrm{CO_{2\text{-}eq}}$ ha$^{-1}$ yr$^{-1}$) from the top peat surface (0–50 cm)
CD = carbon density (t C m$^{-3}$) = BD (t m$^{-3}$) × TOC (%) = 0.14 × 50% = 0.07 t C m$^{-3}$
PVL = peat volume loss (m$^3$ ha$^{-1}$ yr$^{-1}$) = subsidence rate (m yr$^{-1}$) × 10,000 m$^2$
0.6 is a factor of oxidized subsidence (adopted from [8])
3.66 is a conversion factor elemental C to $\mathrm{CO_{2\text{-}eq}}$

Using the equation above and the previously mentioned BD and TOC values, every 1 cm per year subsidence in the drained peat emits 13 to 16 t $\mathrm{CO_{2e}}$ ha$^{-1}$yr$^{-1}$, with an average of 15 t $\mathrm{CO_{2e}}$ ha$^{-1}$yr$^{-1}$. Using a conservative estimate of subsidence rate (see Table 2), ranging 2 to 6 cm per year, carbon loss from drained peatlands is estimated at 30 to 90 t $\mathrm{CO_{2e}}$ ha$^{-1}$yr$^{-1}$. A record of carbon emission from previous studies is compiled in supplementary data (see Table S2). The default value of IPCC carbon emission from oil palm on peat is 40 t $\mathrm{CO_{2e}}$ ha$^{-1}$yr$^{-1}$ [70]. Table S3 summarizes heterotrophic $\mathrm{CO_2}$ emissions from different land-use systems of peatland in Indonesia.

Table 2 presents selected data of subsidence rates in tropical peat of Sumatra and Kalimantan. The range of subsidence rate is conservatively between 2 to 6 cm per year. This data further supports the previous studies that drainage intensity persistently causes peat subsidence [96].

**Table 2.** Subsidence rates under different land use systems in SE Asian drained peatland.

| No | Subsidence Class | Subsidence Rate (cm yr$^{-1}$) | Period of Drainage (yrs) | C Emission (Mg CO$_{2e}$ ha$^{-1}$ yr$^{-1}$) | | Land Use | Location | Reference |
|---|---|---|---|---|---|---|---|---|
| | | | | Published C Emission | This Study | | | |
| 1 | Low (0.3–1.99 cm yr$^{-1}$) | 1.10 | 15.00 | 4.34 | 16.91 | Community oil palm plantation | Aceh Province | [97] |
| 2 | Low (0.3–1.99 cm yr$^{-1}$) | 1.20 | 15.00 | 2.39 | 18.45 | Community rubber plantation | Aceh Province | [97] |
| 3 | Low (0.3–1.99 cm yr$^{-1}$) | 2.00 | 20.00 | 26.50 | 30.74 | Agriculture | Western Johor, Malaysia | [8] |
| 4 | Low (0.3–1.99 cm yr$^{-1}$) | 0.40 | 36.00 | - | 6.15 | Agriculture | Kalampangan, Central Kalimantan | [98] |
| 5 | Low (0.3–1.99 cm yr$^{-1}$) | 0.36 | 12.00 | - | 5.53 | Community oil palm plantation | Hampangan, Central Kalimantan | [98] |
| 6 | Moderate (2.0–3.99 cm yr$^{-1}$) | 3.90 | 5.00 | 62.81 | 59.95 | Oil palm | Indonesia | [10] |
| 7 | Moderate (2.0–3.99 cm yr$^{-1}$) | 3.70 | 19.00 | 59.58 | 56.88 | Oil palm | Indonesia | [10] |
| 8 | Moderate (2.0–3.99 cm yr$^{-1}$) | 3.90 | 12.00 | 64.42 | 59.95 | Oil palm | Kampar Peninsula, Riau, Sumatra | [99] |
| 9 | Moderate (2.0–3.99 cm yr$^{-1}$) | 3.70 | 12.00 | 58.19 | 56.88 | Oil palm | Kampar Peninsula, Riau, Sumatra | [99] |
| 10 | Moderate (2.0–3.99 cm yr$^{-1}$) | 2.80 | 15.00 | 5.82 | 43.04 | Community rubber plantation | Aceh Province | [97] |
| 11 | Moderate (2.0–3.99 cm yr$^{-1}$) | 2.60 | 30.00 | 75.00 | 39.97 | Community rubber plantation | Tanjung Jabung Barat, Jambi Province | [72] |
| 12 | Moderate (2.0–3.99 cm yr$^{-1}$) | 2.40 | 20.00 | 71.00 | 36.89 | Mixed agriculture | Tanjung Jabung Barat, Jambi Province | [72] |
| 13 | Moderate (2.0–3.99 cm yr$^{-1}$) | 2.80 | 40.00 | 85.00 | 43.04 | Mixed agriculture | Tanjung Jabung Barat, Jambi Province | [72] |
| 14 | Moderate (2.0–3.99 cm yr$^{-1}$) | 2.20 | 10.00 | - | 33.82 | All agricultural drained peat | Southeast Asia | [63] |
| 15 | Moderate (2.0–3.99 cm yr$^{-1}$) | 3.21 | 6.00 | - | 49.34 | Agriculture | Misik, Central Kalimantan | [98] |

**Table 2.** *Cont.*

| No | Subsidence Class | Subsidence Rate (cm yr$^{-1}$) | Period of Drainage (yrs) | C Emission (Mg CO$_{2e}$ ha$^{-1}$ yr$^{-1}$) | | Land Use | Location | Reference |
|---|---|---|---|---|---|---|---|---|
| | | | | Published C Emission | This Study | | | |
| 16 | High (4.0–5.99 cm yr$^{-1}$) | 4.30 | 24.00 | 80.00 | 66.10 | Acacia plantation | Riau, Sumatra | [64] |
| 17 | High (4.0–5.99 cm yr$^{-1}$) | 5.00 | 6.00 | 74.48 | 76.86 | Acacia plantation | Indonesia | [10] |
| 18 | High (4.0–5.99 cm yr$^{-1}$) | 5.00 | 14.00 | 74.30 | 76.86 | Acacia plantation | Kampar Peninsula, Riau, Sumatra | [99] |
| 19 | High (4.0–5.99 cm yr$^{-1}$) | 4.80 | 15.00 | 25.23 | 73.79 | Community oil palm plantation | Aceh Province | [97] |
| 20 | High (4.0–5.99 cm yr$^{-1}$) | 5.90 | 20.00 | 53.20 | 90.69 | Community rubber plantation | Central Kalimantan | [56] |
| 21 | High (4.0–5.99 cm yr$^{-1}$) | 5.00 | 25.00 | 76.00 | 76.86 | Acacia and oil palm | Riau and Jambi | [99] |
| 22 | High (4.0–5.99 cm yr$^{-1}$) | 5.27 | 6.00 | - | 81.01 | Community oil palm plantation | Misik, Central Kalimantan | [98] |
| 23 | Very High (>6.0 cm yr$^{-1}$) | 8.20 | 10.00 | 38.88 | 126.05 | Community oil palm plantation | Aceh Province | [97] |
| 24 | Very High (>6.0 cm yr$^{-1}$) | 9.20 | 10.00 | 40.64 | 141.42 | Community oil palm plantation | Aceh Province | [97] |
| 25 | Very High (>6.0 cm yr$^{-1}$) | 8.20 | 1.00 | 48.09 | 126.05 | Community oil palm plantation | Aceh Province | [97] |
| | Min | 0.36 | 1.00 | 2.39 | 5.53 | | | |
| | Max | 9.20 | 40.00 | 85.00 | 141.42 | | | |
| | Mean | 3.89 | 15.92 | 51.29 | 59.73 | | | |

Drainage and land use disturbance on peat cause peat subsidence everywhere in this world [59,63,68,100–103]. In the first year of drainage, the rate of peat subsidence is very high. Then, the peat subsidence rate is stabilized in the long term, for example, five years post-drainage construction [8]. The rate of subsidence in the drained peat is continuous; hence, carbon emission persistently occurs. Rewetting the drained peat and the subsequent water management would reduce the subsidence rate at about 20–30% [1,64]. This suggests peat oxidation in the drained peat causes a large amount of carbon emission. The availability of oxygen in the top peat layer above the groundwater table leads to significant microbial activities that decompose organic compounds in peats [54,104,105]. Peat subsidence in converted and drained peats is irreversible and endless.

## 9. Assessment of Peat Subsidence and C Emission

The rate of peat subsidence is varied (See Table 2). On average, post 16 years of drainage, the average rate of subsidence is 3.9 cm yr$^{-1}$. The amount of C emission reported in literature ranged from 2.39 to 85.00, with an average of 51.29 Mg CO$_2$e ha$^{-1}$ yr$^{-1}$. In general, the reported C emission is relatively close to the 2013 IPCC default values of the CO$_2$e emission factor in drained tropical peats, ranging from 5.5 to 73 Mg CO$_2$e ha$^{-1}$ yr$^{-1}$. Moreover, using Equation (1) above, the estimated C emission in this study ranged from 5.53 to 141.42 Mg CO$_2$e ha$^{-1}$ yr$^{-1}$, with an average of 59.73 Mg CO$_2$e ha$^{-1}$ yr$^{-1}$. Figure 7 presents scatter plots between published C emission based on subsidence rates and the calculated C emission based on Equation (1). The calculated C emission in this review is linear with a subsidence rate. The estimated C emission caused by 1 cm peat subsidence is about 15.4 Mg CO$_2$e ha$^{-1}$ yr$^{-1}$. On average, the estimated C emission from drained tropical peat in Indonesia is 60 Mg CO$_2$e ha$^{-1}$ yr$^{-1}$.

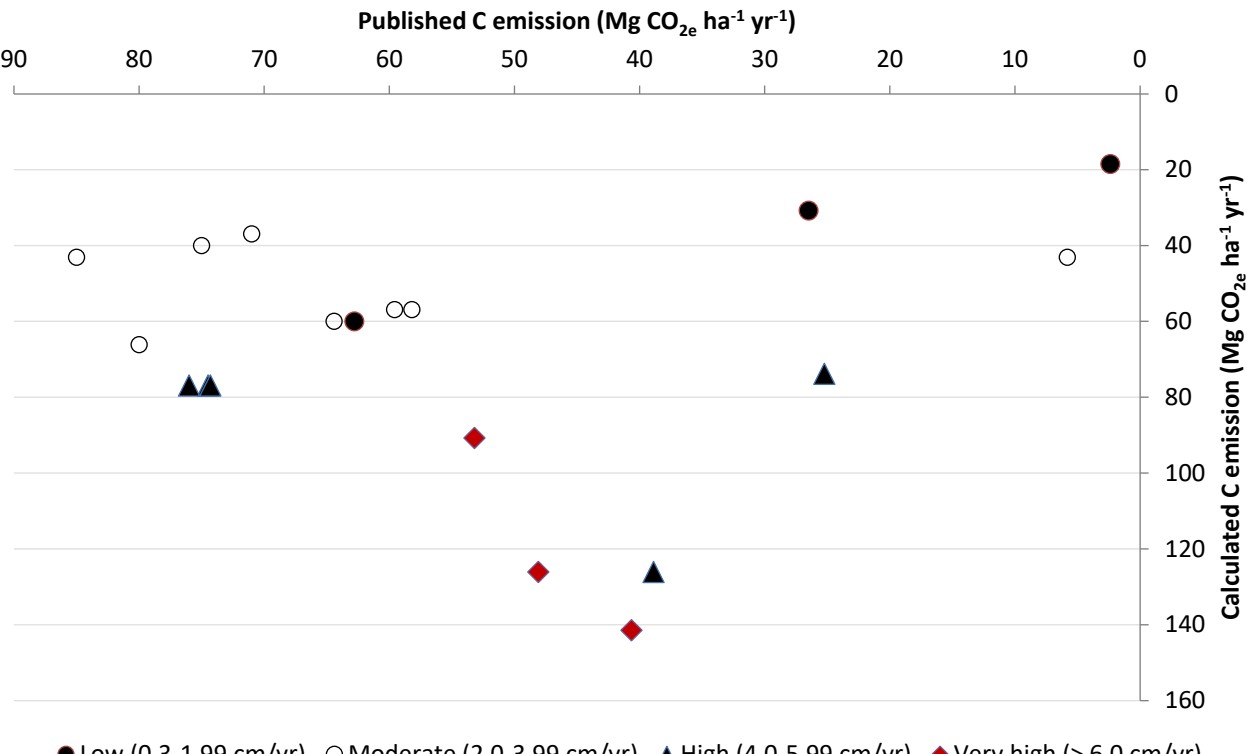

**Figure 7.** A scatter plot between subsidence rate and published C emission data and calculated C emission based on subsidence rates.

## 10. Policy Implication

Despite complex issues and views on peat degradation, ecological degradation of tropical peat forests consists of hydrological change and alteration of peat soil properties. Analysis of ex-ante and ex-post water balance is required to understand the natures of hydrological changes, which then requires a long record of selected climate data, i.e., precipitation, air temperatures, wind, and humidity, which is, unfortunately, commonly unavailable. Field measurements of hydrological properties such as evaporation, hydraulic conductivity, water table height, and water permeability are recommended. Next, selected peat properties, such as BD, TOC, and TN, are important parameters to estimate carbon stock. To manage land productivity, several soil fertility variables, such as pH and nutrients, should be frequently monitored. To sum, it may not be appropriate to assess peatland degradation based only on the 40 cm groundwater table, the exposure of sulfidic materials into acid sulfate soil, and the exposure of quartz sand alone. These criteria are stipulated in the current Indonesian government regulation (See Government Regulation No. 16/2017 on Technical Guideline for Peatland Restoration). Figure 8 summarizes factors that govern tropical peat forest degradation in Indonesia.

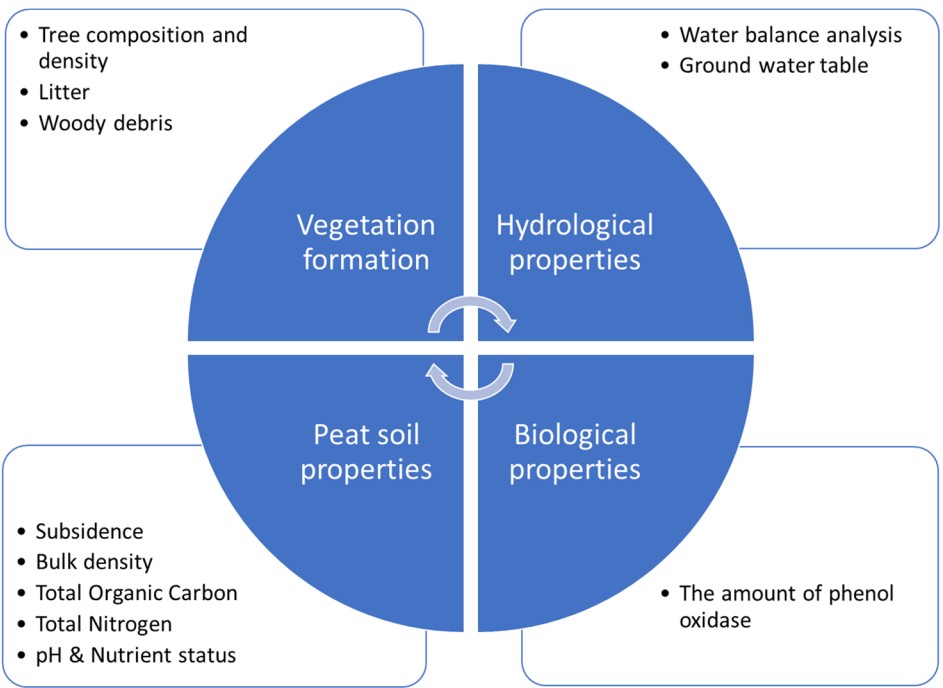

**Figure 8.** Factors that control tropical peat forest degradation in Indonesia.

The current policy on the tropical peat ecosystem in Indonesia focuses on the monitoring of groundwater tables. It is mandatory for all oil palm and timber concession in Indonesia to control the height of the groundwater table at 40 cm all year. In reality, the height of the groundwater table significantly drops, even beyond 100 cm, during prolonged dry seasons associated with extreme years of El Niño. The importance of a high-water table for controlling peat fire is not arguable. Furthermore, high groundwater table reduces the subsidence rates [98]. A waterlogged environment also might reduce carbon emission, despite an increase in $CH_4$ emission. In addition to good water management for keeping a high groundwater table, the subsidence rate measurement is substantially practical to estimate the amounts of carbon emission because the rate of subsidence directly indicates the amount of carbon emission. The contribution of chemical decomposition to subsidence is conservatively about 60% [8]. A high concentration of aromatic organic compounds slows the oxidative rate of tropical peat decomposition [21,106].

## 11. Conclusion and Recommendation

This paper reports that drainage causes an increase in bulk density and TOC decline in the drained upper peat layer, which creates a favorable condition for peat decomposition. Both bulk density and TOC in the undrained peat forest do not alter either in the upper aerobic or inundated bottom peat layers. The use of a groundwater table to estimate carbon loss in drained peat is commonly practiced. Nevertheless, the groundwater table does not always positively correlate with carbon emission measured with the closed chamber. This paper proposes subsidence as a robust and straightforward calculation of carbon emission from drained peats used for all agricultures. Therefore, the Indonesian government and other tropical countries that have peats should consider subsidence as an alternative approach to the groundwater table to estimate carbon loss in drained peat.

**Supplementary Materials:** The following are available online at https://www.mdpi.com/article/10.3390/f12060732/s1, Table S1: Bulk density (BD) and total organic carbon (TOC) in the undrained peat forests; Table S2: Bulk density (BD) and total organic carbon (TOC) in the drained peat forests.Table S3: Total and heterotrophic $CO_2$ emissions from different land use systems of peatland in Indonesia.

**Author Contributions:** G.Z.A., E.G. contribute for writing and original draft preparation, editing and review. N.N. contributes for conceptualization, editing, review and funding acquisition. All authors have read and agreed to the published version of the manuscript.

**Funding:** This research was funded by NORAD, grant number GLO-4251 QZA-16/0172. The APC was funded by NORAD.

**Institutional Review Board Statement:** No applicable.

**Informed Consent Statement:** Not applicable.

**Data Availability Statement:** All data are available in the article and supplementary materials.

**Acknowledgments:** Feedbacks from both Fahmuddin Agus and DIvan Titaley are gratefully acknowledged.

**Conflicts of Interest:** The authors declare no conflict of interest. The funders had no role in the design of the study; in the collection, analyses, or interpretation of data; in the writing of the manuscript, or in the decision to publish the results.

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
