# Peer review of "The Use of Subsidence to Estimate Carbon Loss from Deforested and Drained Tropical Peatlands in Indonesia"

_forests, doi:10.3390/f12060732_

Round 1

Reviewer 1 Report

The Review paper on the title ofThe use of subsidence to estimate carbon loss from deforested and drained tropical peatlands in Indonesia” has dealt with a very important issue of estimating C loss from peatlands in Indonesia. In the manuscript, the groundwater table and subsidence methods of C loss are compared. The use of unpublished data indicates that the validation of the subsidence method is inadequate.

Comments:

-Ground water should be one-word "groundwater". Fix these typos throughout the manuscript.

-Please, don't start any sentence with an abbreviation. Remove these problems from throughout the manuscript.

Line 25: Throughout the manuscript, carbon emission can be termed as “C emission” after having first time defined. The unit needs a space t CO2e ha-1 yr-1

The keywords should be arranged alphabetically.

Line 37: Define organic matters in the first time use, then use OM replacing organic matter.

Line 38: Measurement of groundwater table?

Line 73: total organic carbon also used before so define and abbreviate at first-time use.

Line 74: At 50 cm peat depth interval, how many samples were measured to represent the sample. Was there any explanation in the study report from where the data were extracted?

Line 80 Page 2: arctic?

Line 124: Remove full stop. I can see an additional full stop.

Line 134: Organic matter term is used many times. So, define it at first-time use and then use the abbreviation. # organic matter (OM)

Line 166: unit should be mg C g-1

Line 293-295: Rephrase the sentence.

Line 301: in the range of 2 to 4 m?

Line 305: at each fire event?

Line 366: BD and SOC already defined before!!

The sentence should start as "Bulk density, TOC and peat subsidence (PS) ....

Some recent references are absent in the manuscript.

I would suggest comparing the measurement of subsidence and C loss with the real-time data. There are some works already published. So, insert a paragraph on “Estimates vs real-time C loss data comparison”.

Author Response

Dear Reviewer

Thank you for giving us the opportunity to submit a revised draft of the manuscript “ The use of subsidence to estimate carbon loss from deforested and drained tropical peatlands in Indonesia” for publication in the Special Issue Forest Policy and Global Environmental Governance.  We appreciate the time and efforts that you and three reviewers dedicated to providing feedback as well as valuable insight to improve our manuscript. We have incorporated all of your suggestions, and the changes can be found in the revised manuscript with tracked changes. We add more information on methodology  and discussion sections, revise editorial suggestions, add a figure to show a comparison between published and predicted estimate of  CO2 emissions, and insert photos of subsidence measurement in sever land use types.

In addition, please see the matrix attached for a point by point response to your comments and concerns.

Sincerely, 

The authors,

Reviewer 2 Report

A good contribution.

Our comments contain mostly the support and some suggestions.

Lines 16-17

Lowering water table escalates subsidence, and consequently, carbon is emitted…

A link between subsidence of the soil and carbon emission is not so obvious, as you have stated.

Lines 22-24

To estimate carbon emission from the top layer (0-50 cm) in drained peats, we suggest using BD value range between 23 0.12 to 0.15 g cm-3, TOC value of 50%, and a 60% conservatively oxidative correction factor.

The phrase “we suggest using” is unclear and does not well connected with the following text of the sentence.

Lines 58-77

The text is methodological. Where is your methodology section?

Lines 372-373

Line 372 “to estimate”, line 373 “can be estimated”. Diversify your choice of words.

Line 374

Please use a sign of multiplication instead of the letter “x” in formula (1)

Lines 391-398

It would be much better to present the graphics where to show the dynamics of soil surface subsidence. This approach will give an opportunity to compare your data and the data of other researchers you criticize in the Introduction section. Please provide a curve for your data, and a curve for the data of colleagues, and you will persuade a reader to become your ally.

In the section “Policy Implication” you criticize the current regulations. Yes, some documents of such type can be inconsistent. So please force your reader totally agree with your assessment. Your text is not convincing enough now.

Line 448

This paper briefly shows that…

You has submitted your paper to the high rating academic journal. It is not a place for the brief statements.

Table S1 contains a huge amount of data, but this material has been linked modestly only ones in the line 364.

The manuscript is a review. At the same time, there are author’s data, formula and calculations in the text. Moreover, the tables, figures, formula does not link to references, so a strong prejudice appears that some parts of the manuscript are not a true review.

Even though this is a review, it should be useful to present a methodology section in the manuscript. In this section, the authors would be capable to present their research more clearly. The manuscript topic is a comparison. This quantitative comparison suggests an instrument to compare, and it would be good to present this instrument.  

Author Response

Dear Reviewer

Thank you for giving us the opportunity to submit a revised draft of the manuscript “ The use of subsidence to estimate carbon loss from deforested and drained tropical peatlands in Indonesia” for publication in the Special Issue Forest Policy and Global Environmental Governance.  We appreciate the time and efforts that you dedicated to providing feedback as well as valuable insight to improve our manuscript. We have incorporated all of the reviewers suggestions, and the changes can be found in the revised manuscript with tracked changes. In general, we add more information on methodology  and discussion sections, revise editorial suggestions, add a figure to show a comparison between published and predicted estimate of  CO2 emissions, and insert photos of subsidence measurement in several land use types.

In addition, please see the matrix below for a point by point response to your  comments and concerns.

Reviewer 3 Report

Dear Editor/Authors,

I have read the manuscript forests-1220397, entitled "The use of subsidence to estimate carbon loss from deforested and drained tropical peatlands in Indonesia" written by Anshary et collab. The paper present a very interesting problem in tropical wetlands regions, and the authors have made a good work. The paper is well written, and the bibliography used is representative for the above mentioned theme. Some minor suggestion can be found in the *.pdf file attached. Also, some representative photos should be added, in order to reach a larger audience.

Best regards

Author Response

Dear Reviewer

Thank you for giving us the opportunity to submit a revised draft of the manuscript “ The use of subsidence to estimate carbon loss from deforested and drained tropical peatlands in Indonesia” for publication in the Special Issue Forest Policy and Global Environmental Governance.  We appreciate the time and efforts that you dedicated to providing feedback as well as valuable insight to improve our manuscript. We have incorporated all of the reviewers suggestions, and the changes can be found in the revised manuscript with tracked changes. We add more information on methodology  and discussion sections, revise editorial suggestions, add a figure to show a comparison between published and predicted estimate of  CO2 emissions, and insert photos of subsidence measurement in several land use types.

In addition, please see the matrix attached for a point by point response to your comments and concerns.
